

# Students' sense of belonging and its impact on effectively teaching about
# environmental changes in high latitudes during a master's programme
Karoliina Särkelä[1], Janne J. Salovaara[1,2], Veli-Matti Vesterinen[3,4], Joula Siponen[1,2], Katariina Salmela-Aro[3],
Laura Riuttanen[1], Katja Anniina Lauri[1]
[1]Institute for Atmospheric and Earth System Research (INAR), University of Helsinki, Finland
[2]Helsinki Institute of Sustainability Science (HELSUS), University of Helsinki, Finland
[3]Faculty of Educational Sciences, University of Helsinki, Finland
[4]Department of Chemistry, Faculty of Science, University of Turku, Finland
*Correspondence to*: Katja Anniina Lauri (katja.lauri@helsinki.fi)
**Abstract.** Sense of belonging plays a significant role in students' academic success. For the 'Environmental Changes at Higher
Latitudes' master's programme, success is effectively communicating geoscience research and ideas to the students. This study
explores students' perceived sense of belonging, the conditions for belonging among master's students of this particular
programme, and the impact of belonging on educational effectiveness in a climate change context. This programme is organised
jointly between universities of three Nordic nations and for it—and for the multilocality of the geoscience themes—has a
particularly high degree of mobility. Therefore, the programme lacks elements present in a typical higher education experience,
such as on-site attendance in a physically shared space with a relatively stable group of peers and instructors which are thought
significant for the students' feelings of belongingness. Based on 15 interviews, we elaborate on the findings of the students'
motivation, ability and opportunities to belong and on the construct of their perceived belonging. Emerging from this study, these
constructs for sense of belonging consist of the students' sense of familiarity – familiar elements in the place, surroundings and
culture; sense of recognition – recognised by oneself and others as a peer and a member of the knowledge community; and last,
sense of relevance – finding their studies relevant and interesting. Due to the unique set-up of the programme, the study reveals
insight into elements that support the sense of belonging, crucial in such geoscience and climate education and communication
that might lack the typical shared physical space of a programme.
**1 Introduction**
Environmental changes and the discourse surrounding climate change have become ubiquitous in global society. Among the
various approaches aimed at mitigating and adapting to both anticipated and ongoing changes, education has long been proposed
as a seemingly reliable strategy (Anderson, 2012). Education on climate change and sustainability issues is often characterised by
its interdisciplinary and problem-based nature (McCright et al., 2013) to emphasise the development of practicable skills to tackle
global problems. Various approaches can be taken in this endeavour. Climate science education focuses on teaching the scientific
basis of the Earth's climate system and the factors affecting it, focusing on atmospheric, oceanic and terrestrial interactions. It is a
subset of geoscience education, which covers the Earth's physical systems beyond climate. On the other hand, climate science
education is also a subset of climate education, which involves a broad understanding of the climate system, human impacts, and
policy responses, thus addressing also climate change impacts, mitigation, and adaptation. Education with such importance yet
with such demanding dispositions has been the subject of extensive research and development, encompassing pedagogical
methodologies (Perkins et al., 2018), educational outcomes (Monroe et al., 2019), global implementation (Molthan-Hill et al.,
2019) and professional practices (Salovaara and Soini, 2021). Similarly to sustainability, geoscience education as well is thought





to require proper contextualisation—of being engaged with relevant locations (King, 2008). To continue, organising geoscience
education as situated learning would also suggest that such elements as the learning community and development of professional
identity are to be given more attention (Donaldson et al., 2020) and that feelings coming from the exposure to various contexts,
cultures and communities ought to be better managed in geoscience education (Hall et al., 2022; Todd et al., 2023). More generally,
according to Delors et al. (1996), education of people to manage in the rapid changes of 21st century societies, the four pillars of
learning should be considered: first, learning to know; second, learning to do; third, learning to live together; and fourth, learning
to be. The third pillar, inherently connected to sense of belonging and involving the creation of a new spirit based on understanding
and recognising others' history, traditions, and spiritual values, is highlighted as vitally important. However, the research on the
impact and conditions leading to better communication of geoscience in climate education seems to seldomly address a sense of
belonging, which centres many of the aforementioned topics.

Sense of belonging is a fundamental human need (Maslow, 1943), and feeling relatedness to other people is crucial for all human
motivation (Ryan and Deci, 2000). Sense of belonging can be defined as the emotional attachment that individuals feel towards
specific groups, systems or environments (Maestas et al., 2007) and their perception that their personal attributes fit with these
entities (Hagerty and Patusky, 1995). In higher education, a sense of belonging among students is widely acknowledged for its
influence on academic performance and overall success within the university environment. Students with a high sense of belonging
tend to have high motivation and enjoyment in their studies (Pedler et al., 2022), self-worth (Pittman and Richmond, 2007) and
high academic achievement (Edwards et al., 2022, Pittman and Richmond, 2007), both in online and traditional education set-ups
(Edwards et al., 2022; Thomas et al., 2014). Sense of belonging is widely recognised as essential for fostering student engagement
in their studies (Thomas, 2012). While engagement is important across all educational contexts, its significance is particularly
heightened in the training of professionals in climate change, given the urgent nature of the issue.

Previous studies have examined how various domains contribute to students' sense of belonging, including social relationships,
academic environments, physical places and overall surroundings, encompassing the entirety of the higher education experience
(Ahn and Davis, 2020). However, it is evident that sense of belonging remains highly personal, with no one-size-fits-all solution
(Cohen and Viola, 2022), and ultimately, a student's sense of belonging is grounded in their perception of their connection to a
chosen group, place or other entity (Allen et al., 2021; Mahar et al., 2014). Therefore, it is essential to understand the underlying
constructs of belonging, independent of specific contexts. This entails understanding the emotional responses and interpretations
that lead to both high and low feelings of belonging. By uncovering these fundamental elements, we can gain insights into how to
effectively nurture and support students' sense of belonging across diverse educational settings.

This study examines the factors that foster and cultivate a sense of belonging within a particular, interdisciplinary master's
programme on environmental changes at high latitudes; operates across multiple universities; and involves students attending
courses in different countries and institutions. Consequently, the student group is dispersed across various institutes and countries,
diverting from what is typically associated with the graduate student experience, such as on-site attendance in a physically shared
space with a relatively stable group of peers and instructors. Due to the unique set-up of the programme, exploring how students'
sense of belonging evolves and what supports it in these circumstances can bring about insights into what conditions are important
for sense of belonging, even without permanent location, institute or people to attach to, thus seemingly continuously challenging
the students' sense of belonging. How does a sense of belonging evolve in students who are subject to constantly shifting teaching
methods and ever-changing surroundings? In this study, we focus on aspects relevant to sense of belonging, adapting the Allen et





al. (2021) framework to emphasise the students' perceptions of their belongingness. Thus, we ask the following questions: *what*
*conditions support and foster a sense of belonging in climate education?*, and: *what attributes do students perceive in their*
*experiences to affect their belongingness?*

**2 Theoretical background**
Recent re-conceptualisations suggest that belongingness is a dynamic and non-static process that is dependent on situational factors
and that the sense of belonging fluctuates over time (e.g. Guyotte et al., 2019). Rather than focusing merely on social connections,
belongingness is proposed to be seen as a 'situated practice' that is rooted in place (Grawett and Ajjawi, 2022). To continue, *state*
*belonging,* referring to a sense of belonging that fluctuates over time and is context-dependent, is distinguishable from *trait*
*belonging,* referring to an individual's inherent tendency to feel belonging irrespective of context (Allen et al., 2021). In higher
education research, a sense of belonging has been suggested to be composed of feelings associated with various domains, such as
the academic environment and community, institutes, people and places, with their cultural significance (Ahn and Davis, 2020;
Thomas, 2012). The importance of these elements in contributing to students' sense of belonging varies depending on the
individual, thus making belonging a highly personal experience (Viola and Cohen, 2022) contingent upon individuals' perceptions
of their belongingness (Allen et al., 2021). In geoscience education—also as a practice of communicating geoscience and its ideas
further—belongingness has been recognised, predominantly implicitly, as a relevant element. Situated learning has been suggested
as a potential key direction of pedagogical development in geoscience education, which has thematic ties to a sense of belonging
by its suggested three core components: community of practice—relating to for example social belongingness, authentic context—
relating to for example cultural belongingness, and embodiment—relating to for example academic belongingness (Donaldson et
al., 2020).

Numerous factors contribute to shaping higher education students' sense of belonging, including peer relationships, engagement
and activities with and within the academic community, personal well-being and connection to physical and cultural environments
(Ahn and Davis, 2020). Overall, students' sense of belonging in higher education is heavily influenced by the quality of
relationships they form with their peers and faculty members (Thomas, 2012). Recent studies have also highlighted the role of
place and surroundings as key elements in shaping one's belongingness (Abu et al., 2021, Ahn and Davis, 2020). As higher
education becomes increasingly mobile and organised online, belongingness too gets cultivated less in fixed times and spaces
(Grawett and Ajjawi, 2022). The rapid shift to online education due to COVID-19, although improved flexibility and accessibility,
had a trade-off; challenges arose in maintaining and altogether having a lower sense of belonging among students who were no
longer anchored to the physical and temporal boundaries of traditional educational settings (Abu et al., 2021). To continue,
geoscience students can also experience a low sense of confidence, which is tied to poorer academic performance (Heron and
Williams, 2022) and coincidentally relevant to feelings of belonging. Belongingness, thus, is an outcome of a process of complex
experiences in multiple spaces and places (Guyotte et al., 2019, after Braidotti, 2006).

In operationalising the theory, we adopt the framing by Allen et al. (2021) on elements that build belonging. Belongingness requires
an opportunity to belong, such as an available social, cultural and environmental context to interact with; a motivation to belong
to that context; and an ability (necessary resources and skills) to interact with it. However, ultimately, the feeling of belonging is
based on the perception of belonging (Allen et al., 2021). In our analysis, we examine how opportunities, motivation and ability to
belong serve as conditional factors that facilitate the development of perceived belonging (Fig. 1). Our focus is on understanding
the emotional responses and interpretations that contribute to shaping this perception.



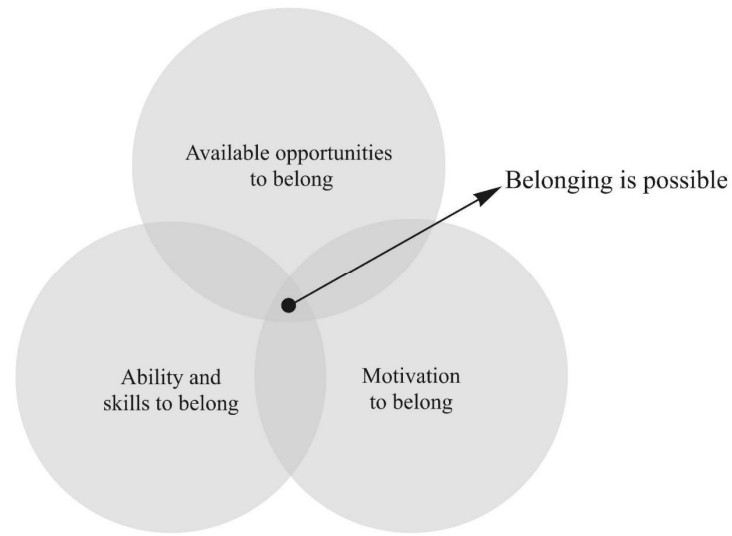

Figure 1. Necessary conditions for fostering belonging, adapted from Allen et al. (2021).

## 3. Materials and methods

### 3.1 Programme

The joint Nordic master programme in Environmental Changes at Higher Latitudes (EnCHiL) is a two-year 120 ECTS Master's programme that is offered by University of Helsinki (UH), Lund University (LU), and the Agricultural University of Iceland (AUI) together with four supporting partner institutes. The first cohort of students started their studies in the autumn of 2020. The programme offers education in multidisciplinary environmental/geosciences with a focus on high latitude ecosystems and societies. The aim is to communicate 'the underlying processes responsible for environmental changes at higher latitudes (Antarctic, Arctic and sub-Arctic areas)' and to educate to the students with a natural science or engineering background a 'deep multidisciplinary knowledge on the past, ongoing and predicted environmental changes at higher latitudes' (The Nordic Master in Environmental Changes at Higher Latitudes, n.d.). In addition to the programme's academic goals, it aims to build a strong Nordic contact network for the students.

Students will study in at least two of the degree-awarding institutes (AUI, UH and LU) in which they are expected to spend a minimum of one semester each (Fig. 2). However, all the students spend the spring semester of their first year at AUI where they study compulsory courses together on campus. In addition to the degree-awarding institutes, students can enrol in courses from the partnering institutes: Aarhus University (Denmark), Greenland Institute for Natural Resources (Greenland), Estonian University of Life Sciences (Estonia) and University of Oulu (Finland).



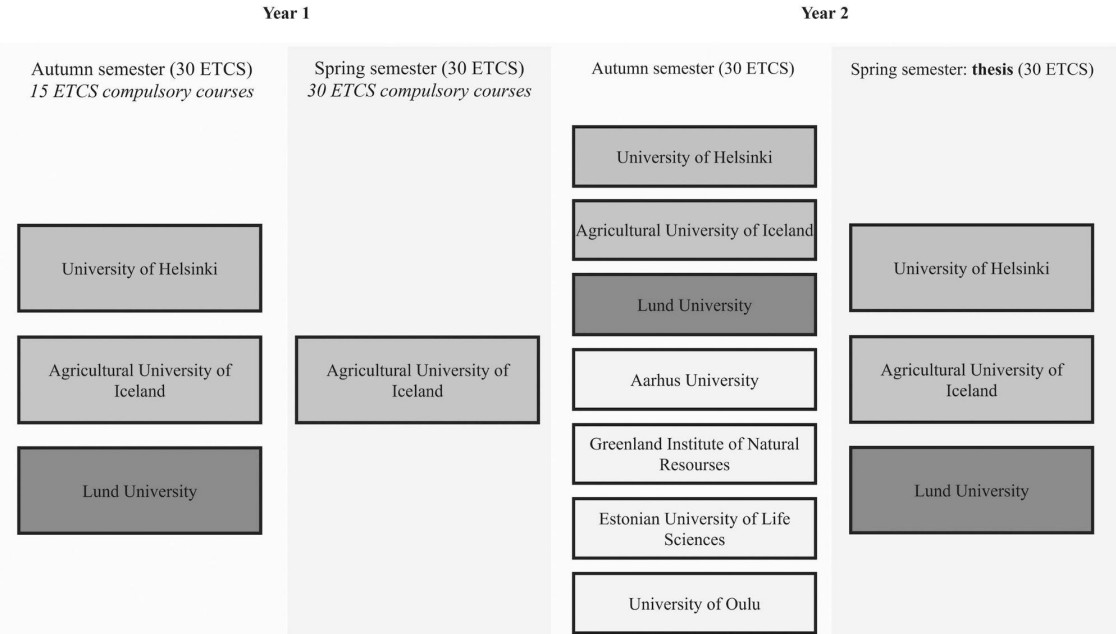

Figure 2. Structure of the EnCHiL programme. Students study the first autumn semester at either the University of Helsinki (UH), Agricultural University of Iceland (AUI) or Lund University (LU). For the first spring semester, all of the students in the cohort study at AUI. During the second autumn semester, the students are free to choose courses from all the degree-awarding and partnering universities and institutions. The second spring semester typically consists of the 30-credit thesis, which the students can submit to any of the degree-awarding universities.

During the first autumn semester of the programme, students start their studies in either AUI, UH or LU. Half of the ECTS credits in the autumn semester are from optional courses, and the other half are from compulsory courses that are offered online for the whole cohort. In the spring, the whole cohort studies in AUI and lives on the campus in Hvanneyri, Iceland. All the spring courses are compulsory. During this semester, the cohort has a field course in Greenland. The second year consists of a 30-ECTS thesis and 30-ECTS optional courses from any of the degree-awarding or partnering institutes. The programme is rather small as the first three cohorts had 5, 10 and 6 students, respectively. Due to the small size of the cohort and the fact that the student body is dispersed among the institutes, only a few students study at the same place at the same time. In the first three cohorts, only 6 out of 21 students were local students from Finland, Iceland and Sweden. Therefore, the majority of the students move from their home country in the beginning of their studies and, due to the mobility scheme, move again to another country at least for one semester. It is to be noted that the main author of this paper is also a graduate of the programme. Thus, this study can be considered to be an insider study (Mercer, 2007) as well, granting the author in question both familiarity to the research case and credibility among the interviewees.

**3.2 Interviews**

We conducted 15 semi-structured interviews with current students and graduates of the programme. The interviewees were approached via direct emails (with messages via phone as reminders) introducing the research themes in general and asking for their interest in participating to this study. Along with the initial email, a Participant Information Letter was sent which also acted as a document of implied consent. The letter informed the participants of the study design, data utilisation, and storage, and explained how their answers and anonymity would be handled in the submission. The primary focus of the interviews was to




explore the sense of belonging that evolved and was constructed during their time in the programme and to deepen our
understanding of the various factors that contribute to shaping the sense of belonging. To continue, we explored the influence of
their peer relationships, staff-student interactions, programme curriculum, personal feelings of achievement and the effect their
physical surroundings might have had on their overall sense of belonging. We were also interested in whether certain courses or
academic experiences held particular significance to their sense of belonging.

The interviewed 15 students were from the first three cohorts of the programme: four that started their studies in 2020, six in 2021
and five in 2022 (see Table 1). The interviews took place in the summer of 2023. Most interviewees had a bachelor's degree in
applied natural or biosciences (e.g. environmental science, geology or biology and related sub-fields), a few had a bachelor's
degree in engineering, a few had a bachelor's degree in fields outside natural sciences.

Table 1. Disciplinary/study backgrounds of the interviewees and the year they started their studies.

| Background / Cohort | 2020 | 2021 | 2022 |
|---|---|---|---|
| Biology | 1 | 1 | |
| Eco/Bio engineering | | 2 | |
| Envrionmental sciences | | 1 | 1 |
| Geology | | 2 | |
| Natural resources | 1 | | 1 |
| Arts/agronomy/geography/other | 2 | | 3 |


At the start of the interview, we elaborated on the concept of sense of belonging through such descriptions as feelings of attachment
to groups, systems or environments, as well as the perception that one's personal characteristics align with those groups, systems
or environments. Participants were then provided with a figure schematising a timeline of their studies, on which they were asked
to visualise how their sense of belonging changed over time. The exercise provided participants with an opportunity to recall and
reflect on their experiences during their studies, and the visualisation served as a reference guiding the conversation through periods
of varying belonging or shifts in their experiences. Thus, the visualisation served as a concise yet comprehensive overview of the
pivotal moments, supporting the verbalisation of their study experience as a whole. The participants had the opportunity to modify
and reflect on their visualisations as they continued their musings throughout the interview.

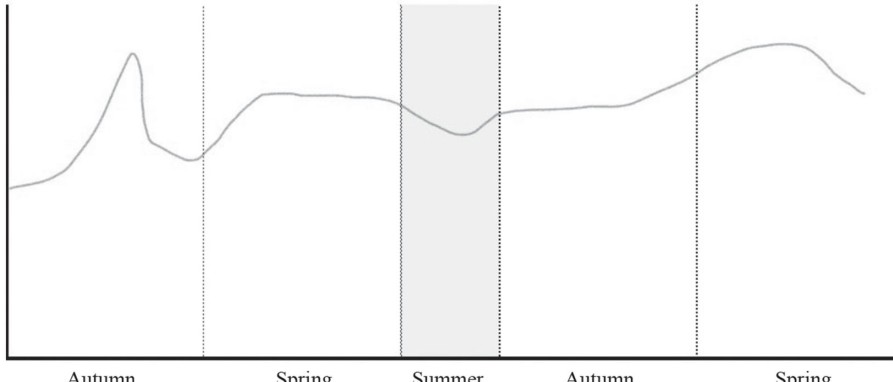


Figure 3. Example draft of the drawing exercise that was used to guide the interview. The interviewees were provided with a
template representing their timeline in the programme. On the template, they could visualise (e.g. by a line) how their sense of
belonging evolved during their studies.




The interviews were conducted as online meetings using the Zoom platform and were 30 to 75 minutes in duration. The recorded
interviews were later transcribed, and these transcriptions were then subjected to content analysis, utilising Atlas.TI for computer-
assisted coding (Bryman and Burgess, 1994). The interviews were initially coded to inductive codes on different reappearances
(Krippendorff, 2018) of the interview's key interests, specifically on perceived degrees of belongingness. These initial codes were
then further grouped into emerging thematic groups. A second round of code grouping was conducted against the formed theoretical
background, as depicted in Fig. 1 of framing inspired predominantly by Allen et al. (2021). The themes emerged as: ability and
skills to belong; motivation to belong; available context to belong to; and emotional responses constructing belonging.

**4 Results and discussion**
The results of our study are presented in two main chapters. First, we focus on the theory-backed analysis, describing our empirical
insight from the interviewees through the previously theoretically conceptualised sense of belonging. Second, we elaborate on an
emerging conceptualisation of sense of belonging, which reveals the construct behind the perceived belongingness rather than
describing the conditions of belongingness.

**4.1 Necessary conditions to belong**
***Motivation to belong***
Belonging is a basic human need, and motivation is what drives the action to fulfil that need (Ryan and Deci, 2000). Most students
identified the motivation to belong as a necessity, especially when related to social interactions within their peer cohort. Sense of
motivation seemed to coalesce with feelings of responsibility and independence; it was their responsibility to seek opportunities to
connect with their peers:

"That [activities they did as a group that brought them closer] was all our ideas. There were no teachers telling us to do that;

you have to also find it in yourself that you want to do this. So maybe it's hard to organise it also". *Interviewee 3*

"So, it's also up to oneself to kind of make that social network around you, and it's not supposed to be the programme's

objective to bring that". *Interviewee 6*

A few interviewees directly expressed a lack of motivation to belong in their cohort. With these cases, lack of belonging was not

explicitly negative, as it was their decision to preclude from such connections:

"Sense of belonging to other people [in the cohort] – was not so strong. I don't think it really disturbs my experience overall

because I already had a group that I belonged to, so I didn't need another group. It didn't disturb me". *Interviewee 11*

"There are some people with ethics that I will not want to associate myself with. So, I do not feel a sense of belonging to

this group". *Interviewee 9*


In both of the previous cases, interviewees stated to have found other groups of people to interact with, and both attached feelings
of belonging to those respective groups. To some extent, the lack of belonging to their peer cohort, even though it was not viewed
as negative *per se*, lowered their sense of attachment to the programme.

***Opportunities to belong***





Sense of belonging is predicated by concrete opportunities to form relationships with groups, systems or environments (Allen et al., 2021). The accessibility of these opportunities was stated as an important factor governing belongingness. Most opportunities that the interviewees recalled were either intrinsically or instrumentally related to conditions that allowed or restricted social interaction. Social interaction has been consistently found to be the most important aspect affecting student belonging (Thomas, 2012).

The students spent their first spring semester in a small rural town called Hvanneyri in Iceland. For many of the interviewees, their stay in a small seclusive place was seen as a catalyst to heightened sense of belonging to their peer group. The small group, with almost all students accommodated at the campus, made them more dependent on each other and thus created a tighter network of the group:

> "[It helped with] that sense of belonging because it's a very small community. Everybody, I mean, my dear, everybody knew almost everybody there and the professors. I mean everything was near, like the houses, the campus and everything. So it was like easier to, you know, talk and communicate. So it helped the sense of belonging to not only to the master's [programme] but to the community, to the campus, to the country". *Interviewee 7*

> "You also need to be more, not friendly, but patient, with other people. Because if we were just a tiny community and you're always with the same people, you don't want to look for trouble. You just want everyone to be happy. You just see the things in a completely different way. I guess in Helsinki you could meet all the time people, so you don't really care about the personal well-being of everyone because there are so many people [...] rather than in Iceland, since we were just a very small community, and there is no one else. You kind of want to know that everyone is feeling great". *Interviewee 12*

For some, the lack of opportunities to interact with other people outside the programme and the small cohort size were viewed as restrictions to social interaction, as interviewees said:

> "I was living in the house with just [the programme] students, so you know, taking the courses together and living together really just kind of sucks you into this one place and makes it, yeah, the lack of opportunities to reach out to new people". *Interviewee 8*

> "I would have preferred living with other people than who I study with, and I would have preferred also living in a big, or just someplace bigger. But that's just how I get energy from outside my study and the stuff I do outside. Then I tap into university, and I bring in energy from outside, and it was very hard in Iceland to get that of course". *Interviewee 6*

The importance of having opportunities to spend time together was frequently brought up. Most interviewees preferred informal interaction in regard to building belonging over more formal interaction, such as during classes. For example, interviewee 3 explained that: 'only school related [student interaction], feels really professional and distant, and then you don't really get the sense of belonging'. Although courses vitally served as spaces for informal interaction to happen, there was also time for non-curricular activities, particularly during residential and field courses:

> "Some courses where we are going on trips, so we have to spend time together outside of studying; also that really helps make you feel belonging". *Interviewee 3*

> "The strongest sense of belonging arises when you are participating in activities outside of the curriculum, that also involve the local students. So in the case of Helsinki, it is for example going to the sauna, experiencing with everyone. There was a strong sense of belonging. In the case of Iceland, it was the impromptu activities we had. We went cross-logging with the other students. We went to the campfire and things like that". *Interviewee 9*



"[A classmate] and I took methods and measurements and the hydrosphere, geophysics [a course]. So we had a couple of
field trips out in the Bay of Helsinki, and so it was very nice to do fieldwork but also to see the city and get to know your
teacher and your classmates a lot more closely. For this reason and definitely after that course, I also felt the belongingness
and in different ways as well from that experience. [...] And you know something like Hyytiälä [a remote forestry research
station], that was of course a way to bond with people. We had time to go to the sauna and to swim and to have like lunch
time with teachers as well". *Interviewee 8*

Altogether, engaging in course peer projects was beneficial for belonging, as interviewee 11 explained: "I think in general, group
work, trying to figure out things together, sort of makes a group. You know, as we did in Hyytiälä, for example. And so the
opposite, I guess you know when you work by yourself, as was my experience mostly in the second year, I was mainly working
by myself. Which probably contributed to not feeling as belonging".

Then again, courses with restricted interaction, mostly mentioned as online courses, were consistently thought of as negative for
belongingness:
"When you don't connect to the people, I think you feel less like you belong. Then you feel like really distant. When I felt
the least like belonging [was] probably when we only had online classes, then we were like all really busy just trying to
understand this and maybe just the only interaction we were having also was related to school. I really didn't feel good. [...]
I almost quit the studies, actually. [...] I was just also feeling really isolated and yeah, but then I still decided to stay".
*Interviewee 3*
"Maybe some of the remote courses that we did kind of in the beginning [decreased the belonging], even though they were
interesting [...] it feels strange to be working online, like doing a group project with people you've never met and you're
just doing it online. And it's very difficult to really kind of connect with the people". *Interviewee 10*

***Ability to belong***
Competency to belong refers to capabilities to connect with other people or environments. Many interviewees brought up how, for
example, mental strain decreased their ability to interact with others or to take part in studies, which then affected their sense of
belonging. Interviewees explained the linkage between personal well-being and belonging:
"If you don't feel good inside, you maybe are not as ready to connect to others. I think, just at least to me, I feel more like
I belong to a group if I feel good myself". *Interviewee 3*
"I just struggled with some [psychological challenges] that haven't been diagnosed. [...] That's just a personal thing that
has been influencing my entire time in this first and second year". *Interviewee 6*

Related to such strain, some interviewees highlighted how courses with a heavier workload, regardless of if they found the topic
interesting, seemed to decrease their sense of belonging. Interviewee 1 said:
"So, it was just a lot of workload, so that's the dip [in the belonging in the chart they drew] because it was just like, you had
been going on for nine months basically and just no break. I mean I was I was working on Christmas Day even. [...] Of
course it was a little bit of a dip [in the belonging] because I thought it was difficult to do my master's thesis. You know, it
took a lot of me to do it".





Similarly, lacking sufficient readiness (e.g. background information) for a course caused stress and struggles with studies, which
again decreased the sense of belonging:
"[Low belonging], like when we were doing the statistics and stuff. Then it really like affected my self-esteem". *Interviewee*
*3*
"Maybe the Greenland course [decreased my sense of belonging] because I don't have a very solid like science background.
So, it was like all these measurements and things like that". *Interviewee 5*

Mental well-being and life satisfaction thus seem to influence the students' sense of belonging, as also noted by Ahn and Davis
(2019). Essentially, fostering a sense of belonging in higher education is not isolated from other aspects of life, as, for some
interviewees, struggles in personal life decreased their capability to belong – to feel belongingness overall. Studies-related stress
can be a significant contributor to students' distress and consequently impair their sense of belonging.
**4.2 Constructs of belonging**
From our analysis of the students' perceived belonging, three dimensions of a *sense of belonging* construct were identified: sense
of familiarity, sense of recognition and sense of relevance. These significantly contributed to the students' feelings of connection,
acceptance and engagement with the academic environment. This construct emerged from the interviews as emotional and/or
perception dimensions that take part in creating belongingness (Fig. 4). These dimensions were associated with a high sense of
belonging in the complex and dynamic educational context that enveloped the students' whole education experience with the
EnCHiL programme.

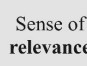

Figure 4. The construct of sense of belonging. Based on the content analysis of the interviewees' experiences, three main
dimensions were identified by the interviewed students: sense of familiarity, sense of relevance and sense of recognition.

***Sense of familiarity***
Sense of familiarity was often associated with various elements supporting belonging. Especially in a new environment,
encountering something familiar created a sense of comfort and ease that assisted the creation of attachment to new places,
countries and institutions. Familiar elements, e.g. in the landscape or in the culture, created a feeling of being 'at home'. Interviewee
2 explained how they found the Icelandic landscape familiar: 'for me, it was like coming home'. Interviewee 13 explained how, at



their apartment in Iceland, they had a mountain view, which reminded them of home: 'I belong to the mountains in [my home country]'. To continue, agriculture-related study activities at the campus were familiar to interviewee 2 because of their background, which therefore supported their sense of belonging:

> "So, I felt like I fit in right away there. [...] So I actually think, from the get-go, I really felt like I belonged. But I think it's a combination of the rurality being very familiar to me and the kind of vibe of the campus being familiar for me as well".

Knowing the local language and culture also made it easier to adapt to new settings. Some interviewees had previously spent time in their new place of residence, which they expressed as helpful for feeling belonging:

> "Well, I mean, for me it was very nice. I already went to Finland that year before Iceland, so I knew the city and I knew, I mean, how everything works". *Interviewee 7*

> "I would say that I had a pretty strong sense of belonging just because I knew a little bit of the area and the language and the culture of people". *Interviewee 8*

Prior knowledge of language and some local customs were important aspects in creating a sense of ease for belongingness, as interviewee 14 elaborated:

> "So, I think the combination of knowing the physical environment, like knowing the almost bureaucratic processes, but also having a community was kind of crucial to the very stabilised sense of belonging".

Familiarity associated with study topics was also important, as interviewee 5 explained:

> "The [course about the] geology in Iceland. [...] So I mean that was something I was very familiar with, so I've had strong belonging there as well".

Familiarity in course topics indicated competency in the topic in question or a connection to their disciplinary identity. Familiarity with study topics was seen as positive also because of emotional attachment to certain topics.

> "With my thesis work, I think it makes me more connected, more enthusiastic about it because this is something that I've seen, and I've walked up on this glacier. You know, I've been there and I've seen this my whole life". *Interviewee 10*

In a study conducted by Kahu et al. (2022), the authors investigated students' sense of belonging during their first year of higher education studies, highlighting the significance of familiarity, particularly during the orientation phase when students acquaint themselves with their studies, surroundings and peers. To continue, according to Antonsich (2010), sense of belonging, marked by a sense of comfort and safety, is highly relevant to one's attachment to a particular place. Given that the programme's students switch their study locations, institutes and social circles at least twice throughout their studies, the importance of familiarity is heightened when they orientate themselves and create a sense of place periodically.

***Sense of relevance***

Perceiving the programme as academically relevant for the student was beneficial for their sense of belonging. Especially courses that resonated with their future goals, hoped career paths and direction in life in general enhanced the experienced belongingness.

> "I did some courses here that I was super happy with, and I was like, OK, I'm on the right side of life. This is what I should be doing. [...] Yeah, I was working on my master's thesis, and I was also doing research [...] , and I just really sense that





this is what I want to do. I want to be out in the field doing some research, making some papers out of the research that I
do, testing things out in nature". *Interviewee 1*
"[When asked which courses increased their sense of belonging] Actually, like the ecosystem ecology [course]. I think
that's the one course that stood out for me because it was something a bit new. It was a big course, and I learned so much,
and that's the lead into what I'm doing today". *Interviewee 5*

Coincidentally, studying courses outside their interest areas lowered the sense of belonging for some students. For some, the
multidisciplinary nature of the programme was challenging as it led to studying topics outside of their main discipline of interest.
For instance, Interviewee 7 described feeling a low sense of belonging during a semester focused on social science subjects, which
were perceived as unrelated to their own disciplinary expertise.
"For me it was like it was very messy like. I was like not fully understanding […] what was the programme about [when]
the second semester was like more like philosophy or environmental values or anthropology or something like that".
Despite integration and a sense of belonging in social relationships, the lack of alignment with their academic interests left them
feeling disconnected from the programme.

One interviewee had doubts in their interest in the programme in general. However, they noted that their sense of belonging
improved when they enrolled in courses more closely related to their personal interests during the spring semester.
"Thinking back, relating to my interest and all, maybe this programme is not the best fit for me because it was maybe more
[focused] into another direction [than the interviewee's interests]. But I guess during the spring, I think there I got the
belonging, doing two courses that are really fitting my interest". *Interviewee 10*

***Sense of recognition***
Recognising oneself as competent and fitting to the study context, and getting external validation for it, was central for supporting
belongingness. Several interviewees emphasised the crucial role of supportive teachers in shaping their positive study experience.
Specifically, validation and recognition from thesis supervisors was significant for fostering a sense of belonging – potentially
because, among all the coursework, thesis work most closely resembles professional research. Interviewee 2 explained that when
they were doing their thesis in a research group, being treated as a colleague made them feel like they fit as a 'scientist' and into
the academic sphere, thus increasing their sense of belonging:
"More and more, so kind of in like discussions around like, "have you looked at this paper", like kind of problem-solving
discussions that I really increasingly feel more and more like a scientist, quote, unquote. So that's growing for me [as a
domain to feel belonging in]".

Interviewee 11 explained how such validation made them feel like they belonged to a research group:
"Having had like a successful thesis project has helped me to you know, get a job. So I feel like I belong to that group now.
[…] You know, doing this stuff well and getting good feedback of course helps in, you know, like establishing yourself in
a group context".

As crucial as it was to be recognised for one's competency by others, it was also important to recognise similarities between oneself
and others alike – all 'fitting in' together:





"[Being around] like my kind of people, basically. Yeah, I think that makes you really belong. Like ok, these are the people
you want to associate with in the future. Through all of my studies […] I always liked the people I met, there's not often
been people that I don't like in this field. […] I mean, like-minded people choose, like, similar paths, basically". *Interviewee*
*1*

Evidently, perceived misfitting then led to feelings of loneliness and alienation. Interviewees expressed feelings of detachment
from their peers as they perceived themselves to be interested in things different than the majority of their peers:

"So, I really felt just on the side and especially because I was doing kind of another thing compared to the others. It was
even harder to feel that I belonged there". *Interviewee 12*

"Feeling of being kind of isolated, you know, because I was doing kinds of different things than, you know, you guys and
the people around me. I felt like sometimes I was a bit isolated, and [that was] of course affecting my sense of belonging".
*Interviewee 10*

Interviewee 10 explained having missed shared interests with others but managed to find people during some elective courses that
better served their interests:

"Of course, I missed sometimes like having a chat about what you're doing that is not just me talking about what I'm doing.
Actually like somebody giving me feedback or having a discussion on like a deeper level related to the interest. But of
course, you know, during the courses I was taking that were really interesting, I had this conversation and I could kind of
have this type of, I don't know, feeling I belonged in a group at a certain time, talking about what we are studying and what
we are learning".


Recognition, both by oneself and others, plays a central role in fostering feelings of belonging. Being recognised for one's
competency by others was also central for the formation of professional identity. For example, Carlone and Johnson (2007) and
Hughes et al. (2021) highlight the importance of recognition in a scientist's identity development. Furthermore, Hazari et al. (2019)
underscore how sense of belonging contributes to disciplinary identity. Sense of belonging and professional identity development
can be thought of as interconnected processes that reinforce each other.

Not being able to share interests or disciplinary identities with other students of the programme was disruptive for some students'
sense of belonging. Even though interdisciplinary education is thought of as essential in addressing the complex issues of climate
change, it also poses challenges to students with strong disciplinary identities and, consequently, to their sense of belonging.
However, some research suggests that exposure to multidisciplinary environments can strengthen disciplinary identities
(Geschwind and Melin, 2016).

During master's studies, one's disciplinary identity is still under process. One interviewee explained how learning and engaging
with the knowledge community affected their sense of belonging, as it seemingly led to the students to create a shared disciplinary
identity:

"I think the more you learn about the topic, the more you feel that you belong there. Because the more you know about the
topic of your studies, just the more you can connect with other people from your programme". *Interviewee 12*





### 4.3 Sense of belonging in climate and geoscience education

Effective climate and geoscience communication strategies in education overlap with elements that relate to the learner's sense of belonging to their learning community, to the cultures of their study contexts, to the field of experts they are developing to be a part of, and to the interactions with the society around them in their future expert role (Donaldson et al., 2020). In addition, factors that foster a sense of belonging among students—such as engaging in deliberative discussions, interacting with scientists, or implementing community projects—are recognised as generally effective in climate education (Monroe et al., 2019). These are elements that contribute to the students' sense of connection and belonging to the educational setting—which in their case encompasses the aforementioned elements and factors. This suggests that effective geoscience communication, also the case of climate change education, is interconnected with students' sense of belonging and vice versa: paying attention to effective pedagogies and methods of communication ought to heighten the students belongingness and their heightened belongingness ought to strengthen the effect of the education. Thus, creating a learning environment where students can connect to the subject matter and its relevant context—be it socially, culturally, contextually—appears as a key element for effective climate change education.

Familiarity and connection with places and locations is beneficial for belongingness. Place attachment can motivate someone to climate action (Devine-Wright, 2013) and is thus a relevant aspect in climate change education. By incorporating local and tangible aspects of climate change and sustainability, educators can foster a deeper connection between students and the subject matter, they could provide a meaningful learning experience while enhancing their understanding of the topic and their sense of belonging—while they could also manage a better comprehension of the plurality of perspectives that are attached to geosciences (Hall et al., 2022).

While interdisciplinary education is essential for addressing the complexity of climate change (McCright et al., 2013) and geoscience education benefits from happening in relevant locations (King, 2008), high mobility and interdisciplinarity can also pose challenges to students' learning and professional identity development (Donaldson et al., 2020; Geschwind and Melin, 2016) and to their sense of belonging. Support for the students' disciplinary and pre-professional identities, in the kind of programmes studied here, is crucial. For example, directing the students to self-determine the scope of their courses can further enhance their sense of belonging, and resulting in a balance with core geoscience concepts and interdisciplinarity of the programme again enhance the communication of the science itself—which too has various relevant disciplinary and non-disciplinary contexts (King, 2008).

### 4.4 Limitations and future research

With our study aimed to shed light on students' sense of belonging within a multidisciplinary master's programme, we will address some acknowledged limitations in contextualising the findings. First, the unique nature of the programme, characterised by its high level of mobility and research orientation, is a notable factor. The alignment between the programme and the future career aspirations of the participants could compare differently for students in other climate science-oriented programmes; thus, the relevance of our conclusions in other educational contexts could also differ. The relatively small size of programme cohorts and particular intensive teaching periods can also influence group dynamics and interpersonal relationships, thereby shaping students' experiences of belonging. This led us to mitigate the limitation by interviewing individuals from different cohorts. Last, the language proficiency of interviewees may have caused potential limitations in their ability to articulate their experiences more effectively during interviews. Despite these limitations, our study contributes valuable insights into the multifaceted nature of sense





of belonging within the context of higher education, particularly in multidisciplinary programmes focused on climate change and
sustainability. With future research, we would address the mentioned limitations in the breadth and width of sample groups, further
mitigating any factors influencing the theories and methods employed here. To continue, future research endeavours on students'
sense of belonging, its effect on transformative learning and epistemic identity development and, foremost, the effect on the
potential of effective impactful geoscience communication and education are surely due.

**5 Conclusions**

The purpose of this study was to explore students' sense of belonging and the conditions for it in a multidisciplinary master's
programme. Our interest in the programme stemmed from its high level of mobility, which poses a challenge to students in forming
a sense of belonging compared to a typical educational setting. The chosen theoretical approach and the formulated framings for
the interviews and further content analysis seemed to function well for the purpose and led to relevant and original insights. The
semi-structured interviews among the purposefully sampled group of 15 students showcased the theory-suggested conditions for
sense of belonging, namely motivation, opportunities and the ability to belong, and their empirical appearance among the students.
Furthermore, an additional construct of sense of belonging emerged from the analysis. This construct consists of the students'
sense of familiarity, recognition and relevance, which, in our view, could help address the sometimes-opaque presence of a sense
of belonging as a vital condition for effectively communicating the concepts and ideas of geoscience in education. Considering
this sense of belonging construct, we thus suggest that educational planning, curriculum design and professional development in
climate change and sustainability-related education in general ought to consider the sense of familiarity, recognition and relevance
as utilisable bridges to strengthen the learners' belonging in a given programme, or context, even if in flux.

**Author contribution:** Conceptualisation and methodology by SK, SJJ, VV-M, and LKA; investigation by SK and SJ; analysis by
SK, SJJ, VV-M, SJ, and LKA; original draft preparation by SK; writing by SK and SJJ; reviewing by SK, SJJ, VV-M, LKA, SJ,
KS-A, and RL; and revision editing by SJJ and LKA.

**Competing interests:** The authors declare that they have no conflict of interest.

**Ethical statement:** This study was performed under ethical guidelines from the University of Helsinki. However, explicit review
by ethical research board for this study was not mandatory.

**Acknowledgements:** The authors wish to thank the interview participants who were willing and open to share their lived
experiences for the purposes of this research. We also thank the board of the Nordic Master in Environmental Changes at Higher
Latitudes, and especially the director of the programme, Prof. Bjarni D. Sigurdsson from the Agricultural University of Iceland for
giving insight into the programme and access to programme documents, and for enabling us to reach the past and present students
of the programme. In addition, we want to acknowledge the editor and editorial office of this journal and the anonymous reviewers
who gave their time and attention to improve the quality of our submission.

**Funding:** This research received funding from The Research Council of Finland, under grant: 340791 (Project: "Learning of the
competencies of effective climate change mitigation and adaptation in the education system") and from the Nordplus programme
of the Nordic Council of Ministers, under grant: NPHE-2023/10209 (Network: "ABS - Atmosphere-Biosphere Studies", Project:



"We belong – contributing to a sustainable development of international education"). Open Access funded by Helsinki University
Library.

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
