# Peer review of "Students' sense of belonging and its impact on effectively teaching"

_EGUsphere, 2025_

## Referee Comment (RC2)

[revised manuscript text omitted]

→ Why does a unique program bring up basic features of something that is so related to context?

→ In a dispersed program, sense of belonging may come from other things compared to a settled program.

→ Some of the Intro refs are limited and/or old. Some Intro claims would require more refs.

→ How does the theory link to the results?

→ I would to be convinced better that the experiences of such a _small_ program are useful to share now.

→ Why are interviews useful as main data collection approach?

→ Coding needs much more attention.
    → Coding process would have fed the results but how is now unclear.

    → The quotes read like random selections---

---

## Author Response (AR1)

Helsinki, 3 September 2025

Dear Editor,

Thank you for your valuable suggestions to improve the manuscript. We have carefully examined them and made changes, which are summarized in the Tables below. Table 1 summarizes Your comments and our point-to-point responses to them, while Table 2 is an update to the responses we gave during the open discussion of the preprint.

Table 1. Point-to-point responses to the Editor's comments.

| # | comments | response |
|---|----------|----------|
| | **Strengthening the Methodological Foundation** | |
| 1 | Consider elaborating on why this particular method is well-suited to your research questions and how it provides unique insights. | We have motivated our methodology and its theoretical background more explicitly in the Theory as well as Materials and methods Sections. On lines 114-118 we tie our approach in the theoretical context with some extra references. On lines 166-171 we motivate the qualitative methodology we have selected. |
| 2 | The coding process, which forms the foundation of your compelling findings, deserves more detailed explanation to help readers understand its rigour and reliability. | We have a more detailed explanation of the coding process in Section 3.2 (lines 204-214). |
| 3 | Moving some of the limitations discussion from section 4.4 into the methodology would provide helpful context about methodological choices. | We have moved the part of Section 4.4 describing our way of mitigating the small size of the programme to Section 3.2. |
| | **Clarifying the Case Study's Contribution** | |
| 4 | Your case study offers valuable insights that could be made even more impactful with clearer positioning. | We have added a positioning clarification in the Introduction (lines 73-81 and 114-118). |
| 5 | Consider expanding the discussion of which findings stem from the unique characteristics of your programme versus those that might apply more broadly to geoscience education. This would help readers better understand the transferability of your results and appreciate what makes geosciences education distinctive in fostering belonging. | We have tried to clarify the broader applicability of our findings in the Results and Discussion. |
| 6 | Strengthening the connection between the programme's specific properties and your theoretical framework would enhance the paper's contribution. | We have tried to clarify this in the beginning of Section 3.2 (lines 166-171). |

**HELSINGIN YLIOPISTO**
**HELSINGFORS UNIVERSITET**
**UNIVERSITY OF HELSINKI**

**Ilmakehätieteiden keskus INAR / Fysiikka**
Matemaattis-luonnontieteellinen tdk
**Institute for Atmospheric and Earth System Research / Physics**
Faculty of Science

PL 64 (Gustaf Hällströmin katu 2), 00014 Helsingin yliopisto
Puhelin +358 2 941 911, https://www.helsinki.fi/en/inar

P.O. Box 64 (Gustaf Hällströmin katu 2), FI-00014 University of Helsinki
Telephone +358 2 941 911, https://www.helsinki.fi/en/inar

| | **Enhancing Organisation and Contemporary Context** | |
|---|---|---|
| 7 | The introduction provides a solid foundation that could be strengthened with additional recent references to position your work within current scholarship. | We have added references to recently published relevant literature. |
| 8 | Consider clarifying the rationale for the current section organisation, particularly the relationship between sections 1, 2, and 4.3, to help readers follow your argument more easily. | As a solution for this, and to better connect Section 4.3 with our results, we have merged previous Sections 4.2 and 4.3. Throughout the manuscript, we have tried to |
| 9 | The conclusions, whilst identifying important patterns, could be developed further to provide more specific insights and recommendations that build on your valuable findings. | We have rephrased and streamlined the conclusions, adding recommendations to potential stakeholders. |
| | **Suggestions for Further Development** | |
| 10 | A small adjustment to have the arrow point to "Belonging is achieved" would better reflect that belonging is possible throughout the intersection of Availability and Ability circles, with Motivation being the activating factor. | We have updated Fig. 1 along this suggestion. |
| 11 | 4.1 provides excellent insights into the social dimensions of belonging. Consider acknowledging that whilst students emphasised social aspects (which is itself an interesting finding), belonging also encompasses connections to science content, academic environments, and geographic settings—dimensions you explore thoughtfully in other sections. This would highlight the comprehensive nature of your belonging framework. | We have now mentioned some other dimensions of belonging and highlighted that in the interviews, the social part rose above others. |
| | The 'Ability to belong' section presents an opportunity to discuss how the programme could support students with diverse learning needs. Consider incorporating perspectives on adaptive education practices and multi-modal learning experiences, which could strengthen your recommendations. Relevant recent work includes:
- Spaeth, E., and Pearson, A. (2023). A reflective analysis on neurodiversity and student wellbeing: conceptualising practical strategies for inclusive practice. J. Perspect. Appl. Acad. Pract. 11, 109–120. doi: 10.56433/jpaap.v11i2.517
- Heron, P.J., F. Crameri, E.F. Canaletti, D. Harrison, S. Hashemi, P. Leigh, S. Narayan, K. Osowski, R. Rantanen, and J.A. Williams (2025), Art, music, and play as a teaching aid: applying creative uses of Universal Design for Learning in a prison science class. Front. Educ. 10:1524007. https://doi.org/10.3389/feduc.2025.1524007 | We have added a paragraph about importance of adaptive education practices under the 'Ability to belong' Subsection. |

Table 2. Point-to-point responses to the Reviewers' comments.

| | Comment | Response |
|---|---|---|
| Reviewer 1 | This paper presents useful findings from a masters program in a unique teaching setup. Based on this unique setup, novel results are presented on understanding how the sense of belonging in students is impacted when a longer-term stable geographic classroom environment is missing. This paper is very well written, presents the results clearly, and overall fits well within the scope of Geoscience Communication. I do not see needs for major revisions. I only have one minor comment to revise and two suggestions, which I don't consider critical and go more into how the results are presented. | Thank you for the positive overall comment! |
| | Figure 1: Shouldn't the arrow point to: "Belonging is achieved" or similar, as Belonging is actually possible on the entire intersection between Availability and Ability circles, and the Motivation finally makes it happen. | That is a relevant point, and we have changed the text in the Figure to "Belonging is achieved". |
| | Section 4.1 is predominantly focussed on the social aspect of belonging. I got the feeling that belonging towards the science content, or the academic and geographic environment, falls a little short here. Maybe it could just be mentioned that most of the students' answers go in the social direction, so that it is clear to the reader that there's more to consider (as is nicely mentioned in other sections). As is also pointed out, belonging is increasing with getting to know something better, which, I guess, can be applied not only to people, but also the study environment, the science content, etc. | Yes, it is worthwhile to specify the focus areas. However, we feel that while it is true that the data we collected has limitations showing belonging to the science content, the academic and geographic environments were rather strongly present in the students' interviews. As a solution, we have added a paragraph in the beginning of Section 4.1 describing the general focus of the interview answers. |
| | Regarding the 'Ability to belong' section: Would it be worth mentioning the barriers for students with more complex learning needs. It is not clear whether there was feedback from students with variable neurotypes, but adaptive education practices to provide multi-modal learning experiences seem important too. e.g.,:
- Spaeth, E., and Pearson, A. (2023). A reflective analysis on neurodiversity and student wellbeing: conceptualising practical strategies for inclusive practice. J. Perspect. Appl. Acad. Pract. 11, 109–120. https://doi.org/10.56433/jpaap.v11i2.517
- Heron, P.J., F. Crameri, E.F. Canaletti, D. Harrison, S. Hashemi, P. Leigh, S. Narayan, K. Osowski, R. Rantanen, and J.A. Williams (2025), Art, music, and play as a teaching aid: applying creative uses of Universal Design for Learning in a prison science class. Front. Educ. 10:1524007. https://doi.org/10.3389/feduc.2025.1524007 | This is an important issue. We did not specifically study or take into account variable neurotypes, but this is an issue that is definitely present, and also indirectly expressed in some of the interviewees' answers. We have added a paragraph about this in the Ability to belong under Section 4.1. |

| | | |
|---|---|---|
| Reviewer 2 | The topic of this study is interesting, as it highlights the importance of issues beyond the content and didactics of programs when understanding study success. Having said that, I am not sure that this study is as useful as it claims to be. Partially, that has to do with my uncertainty on what the study actually claims (showing the issue can be studied, showing how belonging affects study success, showing how to engage with promoting belonging?). Partially, that has to do with the properties of the study (setup, reach, case specificity). | Thank you for your valuable comments. We believe that this study is particularly useful for organisers of joint- and double-degree education programmes. We have tried to clarify this in the abstract and introduction. After sharing the preprint we have actually already received positive feedback from a few joint-degree programme directors and coordinators. Our aim has not been to show how sense of belonging affects study success or how to promote belonging. Some of the authors have conducted another recent study where they tried to understand connections between sense of belonging and transformative learning (Siponen et al., Geosci. Comm., accepted, preprint https://egusphere.copernicus.org/preprints/2025/egusphere-2024-4097/), but in this study the main aim has been to find out the elements that create sense of belonging among students in a constantly changing study setting of this particular degree programme - which also makes it unique, but does not mean that the results would not be useful in the curriculum design of other programmes, particularly international joint-degree programmes and other geoscience programmes involving both fieldwork and online studies. Concerning the reach of the study we would like to emphasise that the programme is small: there were a total of 21 students in the first three cohorts of the programme, and thus the 15 interviews covered more than 70% of all the students and graduates of the programme at the time the interviews were carried out. |
| | The Introduction reads well, but seems to lack some references here and there. Some refs seem to be a little old. I am not sure what the criteria were when separating what is now 1, 2, and 4.3, as content seems to be quite similar. How is geosciences education special? | We have tried to better distinguish the different intentions we have, while addressing generally the overall topics of this paper, in different sections of the submission. In addition, we have added and updated some of the utilised references, although some works which are more seminal to the discussion, have remained the same. Among these references, we have highlighted those that are engaged specifically for their relevance and context of geoscience (education). |
| | The method is only mentioned and not defended: why is this a useful and applicable method? Some limitations are mentioned in 4.4, which should have been addressed in the method. With results so depending on the coding (assuming that the items in 4.1 and 4.2 are constructed from coding) that step in the method needs much more attention. Listing a set of quotes makes nice reading, but I find it not so easy to determine the relevance of the findings. | The choice of the qualitative methodology we have used is not only typical to education research, but also useful for the explorative approach we chose for this study. Of course the small size of the programme has its effect too. With 15 informants a quantitative approach would be challenging. We have added in the beginning of Section 3.2 a further clarification and justification of the method we used. |
| | The case study itself is unclear in terms of relevance and applicability. The Intro claims that due to the unique properties the case allows understanding a general issue. The limitations suggest the opposite. The properties of the program are mentioned, but hardly explained in relation to | We want to thank the Reviewer for their astute comment here. As a response, we have tried to be more explicit by our case selection, and its relevance and implications to geoscience education and communication. |

| | | |
|---|---|---|
| | theory and/or results. For example, which findings can be attributed to the case and which are more general? Which results are specific for geosciences? | |
| | The conclusion is very general, as in "we found stuff" and "there is a need to do something in general". I would not even be surprised that is the case, but I would need to know more about the issues I identified above before I accept the general tendency of this conclusion. | We have revised the Conclusions with aims to correspond to this comment, with the note that the majority of the argumentation and evidence is (also structurally) part of the "Results and discussion" chapter, leaving the Conclusion chapter to be quite explanatory and concise. |
| | More smaller notes, here | |

Yours Sincerely
On behalf of the Authors,

Katja Anniina Lauri

---

## Editor Decision (ED1)

**Review Report**

This paper addresses an important and timely topic, examining student belonging in a masters programme with a unique teaching setup that lacks a stable geographic classroom environment. The research tackles a relevant issue for geoscience education, and the paper is well-written with clear presentation of results that fits well within the scope of Geoscience Communication.

**Opportunities for Enhancement**

**Strengthening the Methodological Foundation** The methodology section would benefit from expansion to help readers fully appreciate the approach's value. Consider elaborating on why this particular method is well-suited to your research questions and how it provides unique insights. The coding process, which forms the foundation of your compelling findings, deserves more detailed explanation to help readers understand its rigour and reliability. Moving some of the limitations discussion from section 4.4 into the methodology would provide helpful context about methodological choices. This additional detail would strengthen confidence in your interesting findings.

**Clarifying the Case Study's Contribution** Your case study offers valuable insights that could be made even more impactful with clearer positioning. Consider expanding the discussion of which findings stem from the unique characteristics of your programme versus those that might apply more broadly to geoscience education. This would help readers better understand the transferability of your results and appreciate what makes geosciences education distinctive in fostering belonging. Strengthening the connection between the programme's specific properties and your theoretical framework would enhance the paper's contribution.

**Enhancing Organisation and Contemporary Context** The introduction provides a solid foundation that could be strengthened with additional recent references to position your work within current scholarship. Consider clarifying the rationale for the current section organisation, particularly the relationship between sections 1, 2, and 4.3, to help readers follow your argument more easily. The conclusions, whilst identifying important patterns, could be developed further to provide more specific insights and recommendations that build on your valuable findings.

**Suggestions for Further Development**

**Figure 1 Refinement** A small adjustment to have the arrow point to "Belonging is achieved" would better reflect that belonging is possible throughout the intersection of Availability and Ability circles, with Motivation being the activating factor.

**Expanding the Belonging Framework** Section 4.1 provides excellent insights into the social dimensions of belonging. Consider acknowledging that whilst students emphasised social aspects (which is itself an interesting finding), belonging also

encompasses connections to science content, academic environments, and geographic settings—dimensions you explore thoughtfully in other sections. This would highlight the comprehensive nature of your belonging framework.

**Inclusive Learning Considerations** The 'Ability to belong' section presents an opportunity to discuss how the programme could support students with diverse learning needs. Consider incorporating perspectives on adaptive education practices and multi-modal learning experiences, which could strengthen your recommendations. Relevant recent work includes:

- Spaeth, E., and Pearson, A. (2023). A reflective analysis on neurodiversity and student wellbeing: conceptualising practical strategies for inclusive practice. J. Perspect. Appl. Acad. Pract. 11, 109–120. doi: 10.56433/jpaap.v11i2.517

- Heron, P.J., F. Crameri, E.F. Canaletti, D. Harrison, S. Hashemi, P. Leigh, S. Narayan, K. Osowski, R. Rantanen, and J.A. Williams (2025), Art, music, and play as a teaching aid: applying creative uses of Universal Design for Learning in a prison science class. Front. Educ. 10:1524007. https://doi.org/10.3389/feduc.2025.1524007

**Overall Assessment**

This paper makes a valuable contribution to understanding student belonging in innovative educational contexts. The unique teaching setup provides an excellent opportunity to generate insights relevant to the broader geoscience education community. With some enhancements to methodological detail, clearer positioning of the case study's broader implications, and more specific conclusions, this work could provide important guidance for educators designing programmes that foster student belonging. The research addresses real challenges in geoscience education and offers a solid foundation for advancing our understanding of how students develop connections to their academic communities.